# Visual Prompting via Image Inpainting

**Amir Bar**[* 1,2], **Yossi Gandelsman**[* 1], **Trevor Darrell**[1], **Amir Globerson**[2,3], **Alexei A. Efros**[1]

[1]UC Berkeley        [2]Tel Aviv University        [3]Google Research

## Abstract

How does one adapt a pre-trained visual model to novel downstream tasks *without task-specific finetuning or any model modification*? Inspired by prompting in NLP, this paper investigates *visual prompting*: given input-output image example(s) of a new task at test time and a new input image, the goal is to automatically produce the output image, consistent with the given examples. We show that posing this problem as simple image inpainting – literally just filling in a hole in a concatenated visual prompt image – turns out to be surprisingly effective, provided that the inpainting algorithm has been trained on the right data. We train masked auto-encoders on a new dataset that we curated – 88k unlabeled figures from academic papers sources on Arxiv. We apply visual prompting to these pretrained models and demonstrate results on various downstream image-to-image tasks, including foreground segmentation, single object detection, colorization, edge detection, etc.[1]

## 1 Introduction

In the past few years, self-supervised learning has gained popularity in computer vision and natural language processing (NLP). The growing capacity of modern deep learning models made them prone to overfitting when trained on relatively small labeled datasets. Self-supervised learning provides a solution to this problem by generating "free labels" for any dataset, without the need for manual annotation, addressing the data hunger in these high-capacity deep learning models. However, features learned via self-supervision are not "ready for use" – they typically need to be adapted for a given downstream task by fine-tuning on some labeled dataset. Could this fine-tuning be avoided?

In NLP, prompting [5] has recently emerged as a way to employ a model for a new task without any additional training. A common way of task-prompting for a specific language understanding task at test time is to provide the trained model with an input corresponding to example(s) of the target task together with the query. E.g., typing the following input prompt:

```
Je suis désolé          I'm sorry
J'adore la glace
```

will prompt the model [5] to perform the task of French-to-English translation, returning:
```
I love ice cream
```

Can this idea of test-time task prompting be generalized to the visual domain? That is, instead of the current situation in computer vision, where each trained model serves its predefined task (e.g. segmentation, detection, classification), can we have a single general model that can perform a wide range of user-specified tasks *without any fine-tuning (i.e., weight modification)*?

In this paper we take a step toward this goal by demonstrating that large-capacity image inpainting models, when trained on the right data, can be surprisingly effective tools for *visual prompting*.

---

* Equal contribution.

[1]Project page: `https://yossigandelsman.github.io/visual_prompt`.

36th Conference on Neural Information Processing Systems (NeurIPS 2022).

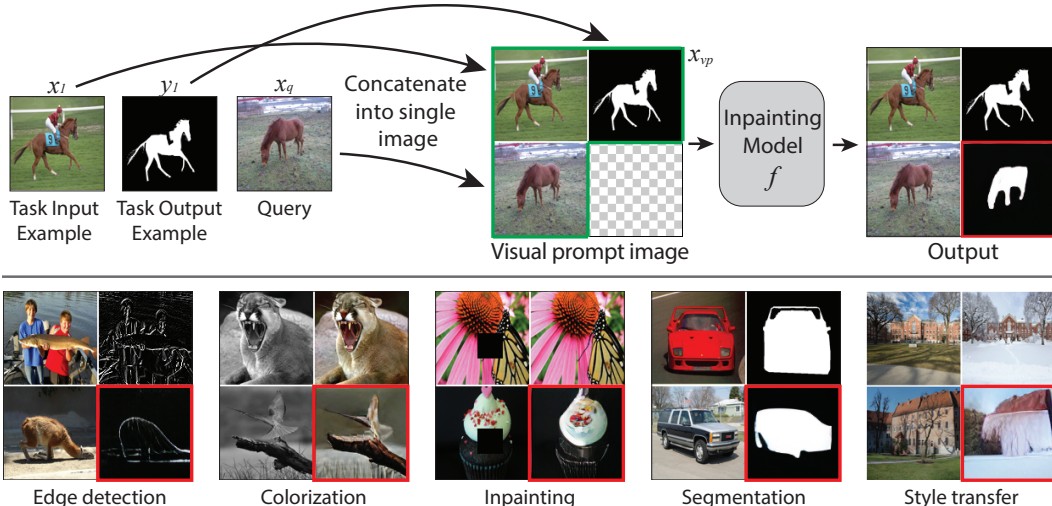

Figure 1: **Visual prompting via Image Inpainting.** *Top*: Prompting Image Inpainting Models. Given input-output example(s) $(x_1, y_1)$ and image query $x_q$, we construct a grid-like single image called a *visual prompt* $x_{vp}$. The visual prompt is composed of the desired task example(s) and a new query image (all in green). The inpainting model goal is then to predict the masked region (red) such that it is consistent with the example(s). *Bottom*: an inpainting model can solve this way various computer vision tasks, given that it was trained on the right data. The model predictions are annotated in red.

As shown in Figure 1, we define each task by constructing a new grid-like image that contains an input-output example(s) of the task and a novel query (green border). The input-output example describes the task, and the image query defines a new input. The model then produces the result by simply inpainting the rest of the image (red border). This setting is most similar to the classic Image Analogies [22] formulation, but is less constrained: instead of explicitly defining the A, A', and B images separately, we simply concatenate them into a single image with a hole (hence, visual prompting is not exactly an analogy since there is no implied left-to-right ordering). Our goal is also not dissimilar from the aims of meta-learning and few-shot learning methods, except that we make no distinction between tasks and example pairs. The only requirement of our formulation is that the tasks must be defined as image-to-image translations, which is a very large subset of vision problems.

To obtain training data that is most useful for our framework, we utilize a domain that spans across a variety of computer vision tasks - figures and infographics from computer vision articles available on Arxiv. We build a large dataset of 88 thousand figures, many of which contain grids of images and their corresponding task results (e.g. images and their segmentation masks/stylized versions/edges, etc.). We then train large-capacity inpainting models to predict randomly masked patches from figures given other patches from the same figure.

Our main contributions are as follows. First, we present a simple yet surprisingly powerful general approach for visual prompting. We show that various computer vision tasks can be treated as grid inpainting problems, given a few examples of task inputs and outputs and a query image. Second, we provide a new dataset that allows a model to learn such grid structures without any labeling, task descriptions, or any additional information about the grid structure. Finally, we show that while using our new dataset for training is essential, adding more generic image data from other sources (e.g. ImageNet) further improves the results.

## 2    Related Work

**Natural Image Inpainting**. Filling empty regions in an image has been widely explored for natural images. Earlier methods used data from the input image itself for inpainting [14, 4, 11, 3, 49], whereas later works utilized datasets of images as source data [19, 38, 56, 30, 31]. Recent methods have attempted to apply transformers to visual synthesis tasks [8, 58, 15, 57, 7]. Due to the exponentially large number of completion options for a single output patch, these approaches rely on a discrete latent codebook [51, 40, 15] which serves as a smaller yet expressive vocabulary. To tackle the multimodal nature of synthesis, different approaches have been proposed to model the distribution over possible

completions [15, 57, 7]. For example, [15, 57] proposed to synthesize images line-by-line using an autoregressive model and [7] have proposed iterative parallel decoding. While the standard inpainting task typically aims to complete blank parts in natural images, our focus is on completing grid-like visual prompts, which require reasoning across multiple images within the visual prompt image.

**Hole-filling as a Pretext task.** Recent work has shown that self-supervised pretraining can generate powerful representations for transfer learning, even outperforming its supervised counterparts on challenging vision benchmarks [37, 18, 9, 21, 6, 10, 35, 16]. Pathak et al. [38] first proposed using hole-filling as a pretext task for self-supervision with Context Encoders, where the goal is to predict an random image region given its context. Based on the recent success of Vision Transformers (ViTs) [13], multiple works have proposed to hole-filling a self-supervised pretext task for ViTs [1, 20, 54]. For example, in MAE [20], the goals is to reconstruct the image given a small subset of input patches. After pretraining on unlabeled data, MAE can produce representations that transfer well when fine-tuned on downstream tasks. Here, we use visual prompting to adapt these models to downstream tasks without any finetuning.

**Few-Shot Learning.** In this setting, the algorithm is trained on a labeled dataset of base classes, from which it should transfer to a set of novel classes given only a few training examples (like 10 or 30) [36, 27, 32, 53, 55, 48, 59, 2]. Unlike Few-Shot approaches, here we do not assume access to a large training set of base-classes, and our architecture is not task-specific. Our approach is Few-Shot only in the sense that we construct a visual prompt that contains one or two task examples.

**Image Analogies.** Hertzmann et. al. [22] proposed the framework of Image Analogies for texture synthesis, where the algorithm is given a pair of training images (A and A') and a query image (B). The goal is to synthesize a new image (B') conditioned on the query, following the relationship inferred from the training pair. Other works have used analogies in style transfer [50], and as a supervised image synthesis task [41]. Predicting the correct completion was previously modeled as a classification problem [43, 23], and other works have explored analogies in the context of learning different transformations between pairs of images [34, 46]. Unlike these approaches, we use inpainting MAE models that learn from data, without assuming any predefined analogies structure.

**Prompting in NLP.** With the recent success of large unsupervised language models [44, 12], Brown et al. [5] presented how a variety of NLP problems can be reformulated to a text completion problem given a predefined prompt, which can be used to solve different tasks without any finetuning. Prompting was shown to be a useful tool for solving various NLP tasks and benchmarks [39, 5]. More recently different approaches to prompting have emerged including Prompt Engineering [5, 33], Prompt Ensembling [25], and Prompt Prefix Tuning [29, 28]. Inspired by the success of prompting in NLP, we aim to study prompting in computer vision where prompting hasn't been widely explored.

## 3   Visual Prompting via Image Inpainting

We turn to describe how to perform visual prompting using Image Inpainting models. In Section 3.1, we describe our proposed inpainting model, which is a combination of MAE and VQ-GAN. We then proceed to discuss visual prompting and propose different ways to create visual prompts in Section 3.2 (see example in Figure 1). Finally, we describe the dataset we collected for training our model in Section 3.3. The training process is illustrated in Figure 2

### 3.1   Inpainting using MAE-VQGAN

Given an input image $x \in \mathbb{R}^{H \times W \times 3}$ and a binary mask $m \in \{0, 1\}^{H \times W}$, the goal of an inpainting function $f$ is to synthesize a new image $y \in \mathbb{R}^{H \times W \times 3}$, with the masked regions filled:

$$y = f(x, m)$$

Figure 2: **MAE-VQGAN Architecture.** During training, an input image is patchified, masked and fed into an MAE [20]. For each masked token, the decoder outputs a distribution over a pretrained VQGAN [15] codebook. The model is trained using cross entropy loss.

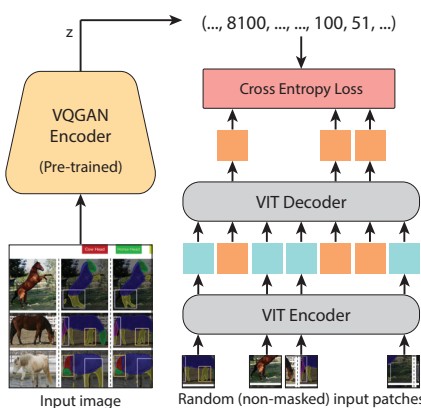

To implement $f$ with a neural network, it is necessary to consider design choices like the network architecture, how to train it, and whether it outputs a distribution over possible completions or pixels. We propose the MAE-VQGAN model, which combines ideas from MAE [20] and VQGAN [15]. Following the design of MAE, the model is based on ViT [52, 13] and it is trained via masked auto-encoding by randomly masking image patches and then applying $f$ to reconstruct the image from the non-masked parts. MAE-VQGAN models the distribution $p_\theta(z_i|x, m)$, where $z_i \in V$ is a visual token from a VQGAN vocabulary $V$ that corresponds to the $i^{th}$ ViT patch. For simplicity, we use a fixed ImageNet pretrained VQGAN codebook.[2] Unlike MAE which directly predicts pixels, MAE-VQGAN assigns probabilities to visual tokens via a softmax layer, which is better suited for capturing ambiguities. During training, we obtain ground truth visual tokens by mapping the image to visual tokens indices using the VQGAN encoder. The model is trained using cross entropy loss.

Let $\hat{z} = (\hat{z}_1, ..., \hat{z}_k)$ be the ordered set of predicted visual tokens. To obtain $\hat{z}_i$, we use argmax:

$$\hat{z}_i = \arg\max_{z_i} p_\theta(z_i|x, m)$$

Then, to decode the visual tokens to pixels, we apply VQGAN decoder to $\hat{z}$ to obtain $y$.

### 3.2 Prompting Inpainting Models

To prompt an inpainting model, we construct a *visual prompt*, a grid-like image composed of task input-output example(s), and a new query image. The model then has to inpaint the rest of the image such that it is consistent with the task defined in the examples (see Figure 1).

Let $S = \{(x_i, y_i)\}_{i=1}^n$ be the set of input-output examples where $x_i$ is an image and $y_i$ is a function of $x_i$ (e.g $y_i$ is a segmentation mask). We assume $n$ is small (one or few examples). Then, given $S$ and a new input query $x_q$, the goal is to predict the corresponding label $y_q$. To prompt the inpainting model discussed in Section 3.1, we need to define a function $g$ that maps the examples set $S$ and query image $x_q$ to a new image and a mask:

$$[x_{vp}, m] = g(S, x_q)$$

The image $x_{vp}$ is the visual prompt and the mask $m$ defines the masked region $f$ has to predict. For a given task, there might exist multiple implementations of $g$ that can be considered. The goal of the inpainting model is to reason about the visual prompt $x_{vp}$, and output a plausible completion without performing any additional training:

$$y_{vp} = f(x_{vp}, m)$$

To obtain $y_q$, we just take the part of $y_{vp}$ corresponding to the mask $m$.

**Visual Prompt Engineering.** For the visual prompting to work, $g$ should output a good visual prompt, composed of the examples $S$ and query image $x_q$. Therefore, $g$ has to determine where and how to embed the inputs in the visual prompt image, considering the nature of the completion task. All the functions $g$ used in this work were hard-coded and manually engineered. In most cases, $g$ stacks the examples and image query horizontally by creating an image grid of $(n + 1) \times 2$ cells, where the $i^{th}$ example is placed in the $i^{th}$ row, and the image query is in the last row. The grid has a fixed size, and therefore before populating it the input-output example pair(s) and query are first resized. Another consideration is how to draw every $(x_i, y_i)$ pair. For example, if $y_i$ is a segmentation mask, we can choose to use different colors to draw it. In Section 4.4, we describe different prompt design choices and their effect on the results.

**Visual Prompt Ensembling.** There could be multiple options to define $g$. The idea in prompt ensembling, inspired by NLP [25, 28], is to construct multiple different prompts, apply the inpainting model $f$ on each prompt individually to obtain a set of predictions. The final prediction can be determined, for example, via majority voting, or weighted average. For simplicity, here we use a simple average.

### 3.3 The Computer Vision Figures Dataset

The images produced by $g$ are by construction not natural. Specifically, these images have a grid-like figure structure that stitches together images coming from different distributions, like natural images and segmentation masks. Therefore, a model trained on a standard dataset (e.g., ImageNet [42]) might struggle to process these grid-like images. To mitigate the domain gap, we collected a new dataset.

---

[2]Using a publicly available checkpoint from https://github.com/CompVis/taming-transformers

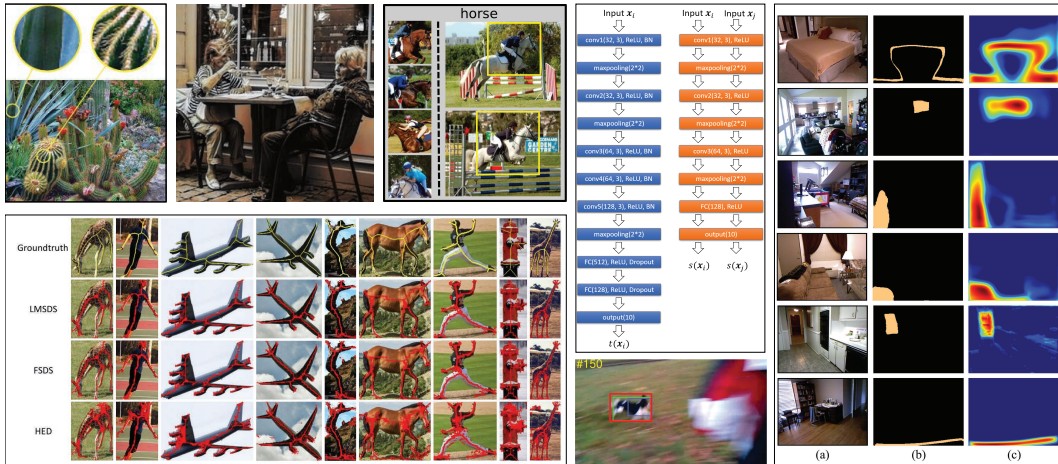

Figure 3: **Random images from our Computer Vision Figures dataset**. We curated a dataset of 88k unlabeled figures from Computer Vision academic papers. During training, we randomly sample crops from these figures, without any additional parsing.

The Computer Vision Figures (Figures) dataset consists of $88,645$ images that more closely resemble the structure of our visual prompts. The dataset was collected from Arxiv, the open-access web archive for scholarly articles from a variety of academic fields. Arxiv sources are publicly available to download starting from 2010. We downloaded all paper sources from 2010 to 2022 and selected the Computer-Vision partition "cs.CV" sources, as they contain images that more closely resemble a grid structure, as shown in Figure 3. To remove unrelated source images like graphs or charts, we manually tagged 2000 images and trained a binary image classifier to assign a high score to source images in a figure-like structure with at least one natural image. We then used the classifier over the entire data to keep only the most informative source images, coming from $23,302$ different papers. We randomly partitioned $90\%$ of the data to train and left the rest for validation. We include a datasheet with more information in the Supplementary Material.

## 4 Experiments and Results

To study visual prompting, we pretrain different models (see Section 4.1) on ImageNet and on the Figures dataset, then quantitatively evaluate the models using different prompts on simple downstream computer vision tasks (see Section 4.2). Using a synthetic dataset, we assess how the choice of model and data affect the success of visual prompting in Section 4.3, and explore different prompting design choices in Section 4.4. We provide a large variate of qualitative results both in this section as well as in the Supplementary Material.

### 4.1 Models and Baselines

To study the effect of model choice on prompting results, we experiment using different models, including MAE-VQGAN (see Section 3.1) and several other inpainting models briefly described below.

**VQGAN** [15] is an autoregressive transformer model used for inpainting and image generation. Visual tokens are predicted sequentially, line-by-line, and the model is trained using cross-entropy loss. The VQGAN model codebook is used to encode visual tokens, and it is trained beforehand using perceptual loss [26] and GAN loss [17]. We train it on ImageNet and our Figures dataset, following hyperparams in [15], and use a pretrained codebook with a vocabulary of size $|V| = 1024$.

**BEiT** [1] is a masked auto-encoder. The model maps each input $16 \times 16$ patch to a visual token from a d-VAE [40] vocabulary of size $8192$. To encode each visual token, the image is first resized to $112 \times 112$ and then mapped to 196 tokens. We use the publicly available BEiT large model, pretrained on ImageNet-21k. We also pretrain a large BEiT model on Figures for 1000 epochs.

**MAE** [20]. Similar to BEiT, MAEs attempt to reconstruct a masked input image. Unlike in BEiT, the model directly regresses pixels and it is trained with $l2$ loss. During pretraining, only non-masked

Table 1: **Visual prompting results on computer vision tasks.** For Foreground Segmentation and Single Object Detection, we report the *mIOU* score. For Colorization, we report the *MSE*.

| Model | Foreground Segmentation ↑ | | | | Single Object Detection ↑ | | | | Colorization ↓ | |
|---|---|---|---|---|---|---|---|---|---|---|
| | Split 0 | Split 1 | Split 2 | Split 3 | Split 1 | Split 2 | Split 3 | Split 4 | MSE | LPIPS |
| Copy | 12.92 | 17.90 | 13.52 | 15.29 | 12.14 | 13.50 | 13.03 | 12.38 | 2.63 | 0.75 |
| BEiT (IN-21k) | 0.38 | 0.93 | 0.90 | 0.95 | 0.24 | 0.32 | 0.19 | 0.10 | 1.25 | 0.73 |
| VQGAN (IN-1k) | 6.96 | 10.55 | 9.59 | 9.43 | 5.19 | 4.99 | 5.09 | 5.10 | 2.44 | 0.66 |
| MAE (IN-1k) | 1.92 | 6.76 | 3.85 | 4.57 | 1.37 | 1.98 | 1.62 | 1.62 | 1.13 | 0.87 |
| MAE-VQGAN (IN-1k) | 2.22 | 7.07 | 5.48 | 6.28 | 3.34 | 3.21 | 2.80 | 2.80 | 3.31 | 0.75 |
| BEiT (Figures) | 5.38 | 3.94 | 3.20 | 3.29 | 0.17 | 0.02 | 0.14 | 0.16 | 0.60 | 0.70 |
| VQGAN (Figures) | 12.56 | 17.51 | 14.27 | 15.06 | 2.27 | 2.37 | 2.48 | 1.99 | 1.50 | 0.56 |
| MAE (Figures) | 17.42 | 25.70 | 18.64 | 16.53 | 5.49 | 4.98 | 5.24 | 5.84 | **0.43** | 0.55 |
| MAE-VQGAN (Figures) | **27.83** | **30.44** | **26.15** | **24.25** | **24.19** | **25.20** | **25.36** | **25.23** | 0.67 | **0.40** |

tokens are fed into the encoder, which results in a faster training time. We use a publicly released checkpoint pretrained on ImageNet, and pretrain another model for 1000 epochs on our dataset.

**Copy Example.** This simple baseline simply replicates the first example label as the output.

**Implementation Details.** All the models we describe are large transformer-based models [52, 13], with patch size $16 \times 16$, embedding dim $1024$, $24$ layers, and $16$ heads. For training, we used a machine with 8 Quadro RTX 6000 GPUs, with a batch size of $48$. The input image size is $224 \times 224$.

## 4.2 Downstream Computer Vision Tasks

We quantitatively evaluate the inpainting models described above on computer vision tasks.

**Visual Prompt.** Given one example pair and a query image, we structure the prompt in the same fashion for all tasks. We construct a grid of $2 \times 2$ sub-images, where the example pair is embedded in the first row, and the query image appears in the bottom left cell. See the example in Figure 1.

**Computer vision tasks.** We evaluate the inpainting models on standard image to image tasks like Foreground Segmentation, Single Object Detection and Colorization.

- **Foreground Segmentation**. The goal is to binary-segment the query image to Foreground and Background. The example is an image and corresponding binary segmentation mask. The query is a new image, and the goal is to complete a corresponding segmentation mask. We use the Pascal-5i [45] dataset, which is comprised of 4 different image splits where every split contains between 346 and 725 images and associated segmentation masks. For each class, the data contains a few image-mask pairs, together with held-out image queries. For every image query, we choose one random example pair. To evaluate, every pixel in the completed image is first mapped to the nearest Foreground or Background color. Finally, we report the mean IOU (mIOU) metric.

- **Single Object Detection**. Similarly to Foreground Segmentation, the goal here is to binary-segment the object that appears in the query image. However, this task is more challenging than Foreground Segmentation because the example mask is obtained from a bounding box which is more coarse than a segmentation mask. We use the Pascal VOC 2012 dataset using images and their associated detection boxes. For simplicity, we use Pascal annotations to include only images with a single object and filter out trivial images that have an object covering more than $50\%$ of the image. We randomly select an example pair and image query of the same object class and repeat the process with $4$ different random seeds. For evaluation, we follow a similar process as in Foreground Segmentation to obtain a binary segmentation mask. Then we keep the connected component with the largest area using morphological operations and draw a bounding box around it. We report the mIOU results.

- **Colorization**. The goal is to map a gray-scale image to a color image. The example pair is a gray-scaled image and the corresponding color image, as shown in Figures 1 and 4. We randomly sampled 1000 example pairs and image query from ImageNet [42] validation set and converted them to gray-scale to obtain gray-scale and color version for each image. We report the MSE loss and LPIPS [60].

**Results.** We include quantitative results in Table 1, and qualitative completion results in Figure 4. Training on the Figures dataset improves the results for most models in all the downstream tasks. MAE-VQGAN outperforms the other models by a large margin for detection and segmentation and

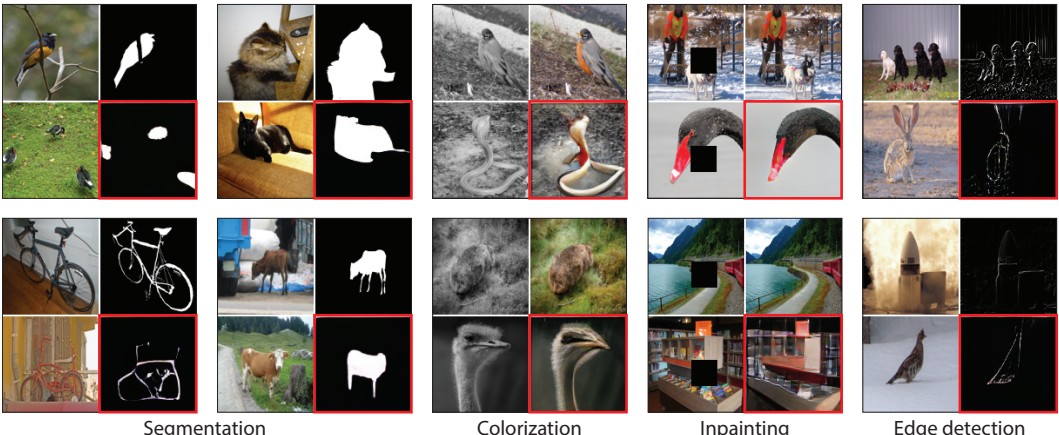

| Segmentation | Colorization | Inpainting | Edge detection |

Figure 4: **Visual prompting prediction examples.** Each visual prompt was fed to an MAE-VQGAN model trained on the Figures dataset. For each visual prompt, the result is marked in red.

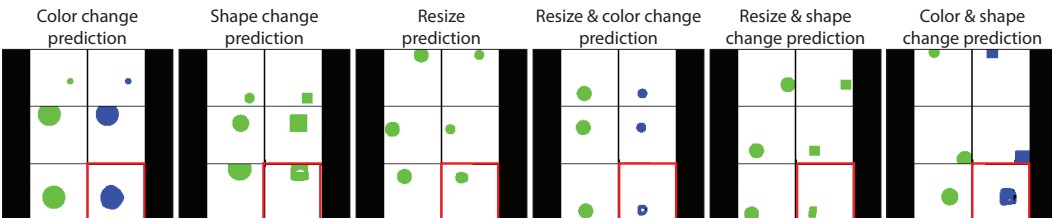

| Color change prediction | Shape change prediction | Resize prediction | Resize & color change prediction | Resize & shape change prediction | Color & shape change prediction |

Figure 5: **Synthetic data study results.** MAE-VQGAN predictions are annotated with a red square.

generates much sharper images than the MAE. We find that VQGAN struggles to output accurate results, likely due to sequential decoding. The BEiT model is outperformed by MAE, most likely because its training process is less sample efficient. For more results, see the Supplementary Material.

## 4.3 Synthetic Data Study

To assess the compositional prediction capabilities of inpainting models, we created a set of 3 simple synthetic tasks and 3 of their combinations, and evaluated each model on 100 examples per task.

**Visual Prompt.** Given two example pairs and a query image, we structure the prompt in the same fashion for all tasks. We construct a grid of $3 \times 2$ sub-images, where the example pairs are embedded in the first two rows, and the query image in the bottom left cell. We include examples in Figure 5.

**Change prediction tasks.** Each example pair is an image of a colored shape, and a corresponding image with an introduced change. The change can be either in color, shape, size or a combination of two changes. Next, we describe each individual task in more detail.

- **Resize**. Each example pair contains an image of a circle, and a corresponding image with the circle smaller in size. The goal is to predict the image with the resized version given image query.

- **Shape**. Here every example pair is an image with circle, and a corresponding image with a rectangle. Both are similar in size and appear in the same location. The goal is to predict the image with rectangle, given a new image query.

- **Color**. Each example pair contains an image of a circle appearing in the same location, with the color changed from green to blue. Given a new image query, the goal is to predict the corresponding image with the circle colored in blue.

**Evaluation.** We map each predicted pixel to its nearest neighbor color from a predefined set of options: black, white, blue, or green. We then measure and report the *color-aware* mIOU, by considering pixel predictions that appear in the ground-truth shape color as foreground and treat the rest as background.

Table 2: **Synthetic data study results.** We report the color-aware mIOU on the six tasks.

|  | Color | Shape | Size | Color & Shape | Color & Size | Shape & Size |
|---|---|---|---|---|---|---|
| Copy | 5.53 | 6.71 | 1.17 | 6.74 | 1.17 | 1.86 |
| VQGAN (IN-1k) | 0.91 | 6.51 | 6.24 | 2.40 | 0.70 | 6.53 |
| BEiT (IN-22k) | 15.99 | 9.08 | 1.26 | 7.23 | 2.84 | 2.66 |
| MAE (IN-1k) | 0.00 | 2.07 | 1.20 | 0.00 | 0.00 | 1.56 |
| MAE-VQGAN (IN-1k) | 0.13 | 2.94 | 3.71 | 0.00 | 0.01 | 3.60 |
| VQGAN (Figures) | 6.96 | 19.11 | 16.21 | 7.40 | 2.24 | 18.41 |
| BEiT (Figures) | 40.92 | 31.43 | 7.12 | **33.10** | **21.21** | 12.98 |
| MAE (Figures) | **70.23** | 43.99 | 34.72 | 19.30 | 18.99 | **46.02** |
| MAE-VQGAN (Figures) | 40.40 | **46.53** | **42.04** | 20.41 | 18.27 | 40.33 |

Table 3: **Comparison to Fine Tuning and Classic 1-Shot Segmentation baselines.** MAE-VQGAN image query and output resolution is $111 \times 111$. CyCTR and FWB resolution is $473 \times 473$ and $512 \times 512$, both approach utilize Pascal 5i labeled baseclasses data.

| Pretraining | # Labeled Images | # Shots | Model | Split 0 | Split 1 | Split 2 | Split 3 |
|---|---|---|---|---|---|---|---|
| Unlabeled ImageNet | 1 | 1 | Finetune MAE | 11.1 | 13.4 | 13.0 | 12.3 |
|  | 4 | 4 |  | 12.9 | 15.8 | 14.3 | 15.0 |
|  | 16 | 16 |  | 13.7 | 16.1 | 16.8 | 17.1 |
| Unlabeled Figures | 1 | 1 | MAE-VQGAN | 32.5 | 33.8 | 32.7 | 27.2 |
| **Labeled** Pascal 5i (Segmentation masks) | $2086 - 5883$ | 1 | FWB [36] | 51.3 | 64.5 | 56.7 | 52.2 |
|  |  | 1 | CyCTR [59] | 67.2 | 71.1 | 57.6 | 59.0 |

**Results.** The results are presented in Table 2, for MAE-VQGAN prediction examples see Figure 5. Without training on the Figures dataset, inpainting models fail to generalize to these previously unseen tasks. The performance of all models improves when they are trained on the Figures dataset. Yet, the same models struggle with combinations of tasks due to the increasing complexity. The VQGAN model utilizes sequential decoding and therefore lacks context, which leads to poor performance. The MAE model outperforms MAE-VQGAN on color, and BEiT performs poorly in size. These models rely on pretrained codebooks (VQGAN and dVAE) that are likely not geared towards these tasks.

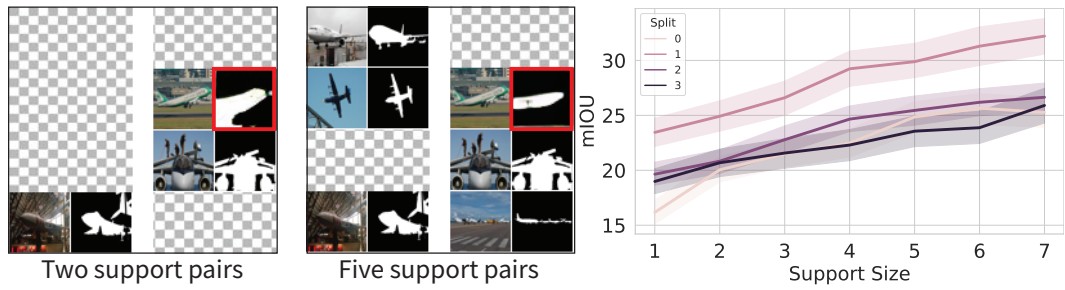

Figure 6: **More examples, better results.** *Left*: we construct visual prompts with increasing number of input-output pair examples, for a fixed query image (inpaintings annotated in red). *Right*: We observe that more examples improve the overall mIOU results on the four Pascal-5i splits.

## 4.4 Analysis

**Comparison to finetuning and Few-Shot baselines.** We include a comparison to baselines that utilize $K = \{1, 4, 16\}$ training examples for each target class. For completeness, we also include the results of FWB [36] and CyCTR [59], classic 1-shot baselines, which we view as an upper-bound of our approach. FWB and CyCTR utilize a fully labeled base classes train set (2086 to 5883 on different Pascal 5i splits). Additionally, their architecture was designed for the foreground segmentation task (e.g, they operate in higher resolution). The results in Table 3 indicate that the Visual Prompting results of MAE-VQGAN trained on Figures are significantly superior to standard

finetuning baselines of MAEs pretrained on unlabeled ImageNet. FWB [36] and CyCTR [59] outperform Visual Prompting, mainly because they pretrain on a large tagged base classes dataset and utilize architectures that are specific to image segmentation.

**Dataset effect.** We evaluate the effect of pretraining on a larger and more diverse dataset. We compare training on ImageNet only, Figures only, and a combination of the two. We report the mIOU results on Pascal 5i for Foreground Segmentation in Figure 7. The MAE-VQGAN trained on ImageNet achieves a consistently low $\sim 5$ points mIOU. The model trained on the combined dataset performs best, which demonstrates that MAE-VQGAN can benefit from additional amounts of unlabeled images.

Figure 7: **Training MAE-VQGAN on more data improves visual prompting results**. Foreground Segmentation results on Pascal-5i, when trained over the Figures dataset and on the combined Figuresand ImageNet dataset.

**More examples, better results.** We study how increasing the number of input-output pair examples in the visual prompt affects the results. Intuitively, we expect that including more examples should reduce ambiguities and lead to better results. We use an MAE-VQGAN pretrained on the Figures dataset, and use data from PASCAL-5i. We construct a large grid that can populate up to 8 examples and an image query. We randomly choose different numbers of examples and randomize the placements. The results in Figure 6 confirm that using more examples leads to better segmentation results.

**Prompt Engineering.** We explore the effect of constructing different visual prompts for Foreground Segmentation and their corresponding MAE-VQGAN results (see Figure 8.a-b). The model generates plausible completions when changing the prompt layout (e.g. horizontal order vs. vertical order) and when changing the mask colors, texture or using only edges (see Figure 9). The mIOU results in Table 4 indicate that the model performs better with a vertical layout and when the segmentation mask colors are black and white. Interestingly, by analyzing the average attention heads of a masked patch token, we observe that the attention changes following the change in the prompts layout (see Figures 8.d-e).

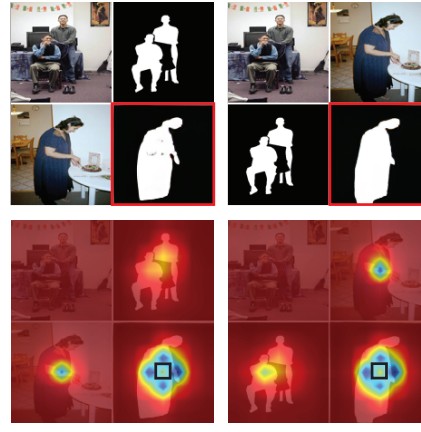

(a) Horizontal Layout      (b) Vertical Layout

**Prompt Ensembling.** Inspired by Prompt Ensembling in NLP [28], given the same example pair and image query, we construct multiple different visual prompts (e.g, horizontal and vertical layouts, see Figure 8.a-b). We then average the completion results. The results in Table 4 on the Synthetic Study tasks demonstrate that utilizing multiple prompts can lead to improved, and more stable performance.

Figure 8: **Prompt layout design.** Two prompt orderings, and the corresponding average attention maps of the selected patch (annotated with black bounding box). The highest attention values appear on similar (corresponding) areas in the query image.

**Style/content extrapolation.** Inspired by the classic example from Tenenbaum and Freeman [47] (on the task originally suggested by Hofstadter [24]), we use MAE-VQGAN to extrapolate letter sequences printed in different fonts (see Figure 10). We find that the model can extrapolate given style and new content (Figure 10a) but that it struggles to extrapolate new content (Figure 10b). The model also struggles to extrapolate more complex letter sequences; the performance deteriorates even if both style and content are given (Figure 10 c-d).

**Limitations.** The focus of this work is to present a proof of concept that shows it is possible to visually prompt simple image inpainting models trained on noisy, unlabeled data. Specifically, we demonstrate how to pre-train a network once, then prompt it to perform reasonably well on many tasks. The fact that this is possible is surprising and scientifically interesting, although this approach is not competitive with supervised task-specific models. For visual prompting to work, the inpainting models require training on the Computer Vision Figures dataset. However, our initial experiments suggest that it can benefit from training on additional natural image data (see Figure 7). Other

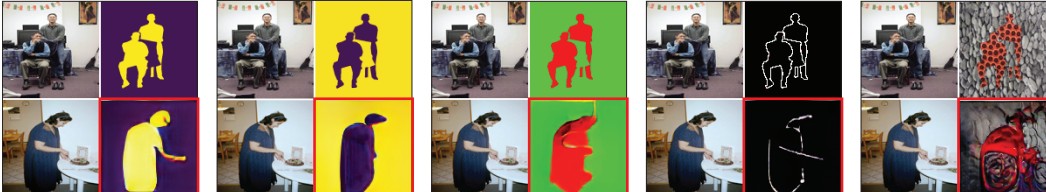

Figure 9: **Task performance under different label choices.** Prompting results when using different mask colors (e.g, purple/yellow vs. green/red), when drawing full mask compared to edges only, and when changing the mask texture. Compared to other alternatives, purple/yellow and black/white (see Figure 8) masks works best.

Table 4: **Prompt Engineering.** Foreground Segmentation mIOU results on Pascal-5i when using different prompt colors.

| | Horizontal | Vertical |
|---|---|---|
| Black/White | 27.17 | 31.57 |
| Purple/Yellow | 23.44 | 28.47 |

Table 5: **Prompt Ensembling.** We report here *color-aware mIOU*. In every line, the result is based on an ensemble of all previous prompts.

| Prompt Layout | Color | Shape | Size |
|---|---|---|---|
| Horizontal | 39.97 | 46.54 | 42.01 |
| + Vertical | 41.31 | 54.71 | 46.18 |
| + Vertical w/ Rows Swap | **44.14** | **60.42** | **49.42** |

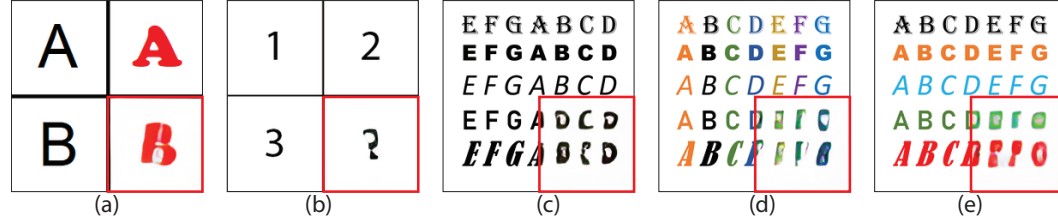

Figure 10: **Style and content extrapolation using MAE-VQGAN.** The model can extrapolate the style of a new content (a), but fails to predict a new content (b). The model struggles to extrapolate new style and content of longer sequences (c-e).

limitations include ambiguities in the task definition, reliance on a pretrained VQGAN decoder, and worse performance when the input-output example(s) are not aligned (see examples in Figure 11).

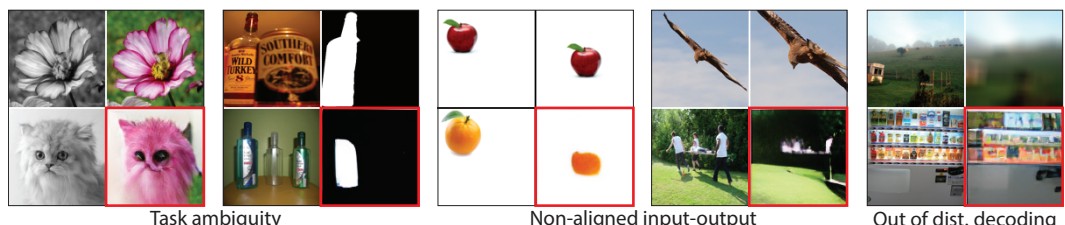

Figure 11: **Limitations and failure cases.** Single input-output example might be ambiguous and can lead to unintended completions. The MAE-VQGAN model performs worse given non-aligned input-output example, and by using a VQGAN vocabulary, it is limited in synthesizing out-of-distribution pixels (like blurry images).

## 5 Discussion

Why does our proposed method, despite its simplicity, perform so well on a large subset of visual tasks? At this point, we do not have a good answer. Clearly, the specific training data we use plays an important role, but the amount of generalization observed is still surprising. Perhaps some of these image-to-image tasks are actually simpler than we believed. But it's also evident that contemporary large-scale inpainting models are learning quite sophisticated long-range co-occurrences and symmetries in the data which can often enable impressive visual reasoning. We hope that our work will encourage further research to better our understanding of what is being learned by inpainting.

**Acknowledgements:** We would like to thank Assaf Shocher for insightful discussions and ideas related to the Figures dataset. We thank Aaron Hertzmann, Sanjay Subramanian, Ofir Press and Ben Bogin for helpful feedback on the manuscript. This project has received funding from the European Research Council (ERC) under the European Unions Horizon 2020 research and innovation programme (grant ERC HOLI 819080). Prof. Darrell's group was supported in part by DoD including DARPA's LwLL and/or SemaFor programs, as well as BAIR's industrial alliance programs. Prof. Efros's group was supported by in part by DoD including DARPA's MCS and/or ONR MURI, as well as funding from SAP.

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
