# OpenReview forum: "Visual Prompting via Image Inpainting"
_NeurIPS.cc/2022/Conference — NeurIPS 2022 Accept_

### Official Review · Reviewer_xQdx · 2022-06-29

**Rating:** 7
**Confidence:** 5
**Soundness:** 3 good
**Presentation:** 2 fair
**Contribution:** 4 excellent

**Summary:**

This paper proposes a uniform framework for various image tasks, such as colorization, edge detection, inpainting, segmentation, and style transfer. This paper treats different image tasks as an inpainting task and achieves good performance. The authors also collect a dataset for training models.

**Questions:**

How did you combine the MAE and VQGAN? During training, did you train VQGAN first? And then use MAE to reconstruct the visual tokens? During inference, do you encode the whole combined image using VQGAN first? And then use MAE to fill the missing part (visual tokens instead of pixels)? And then use VQGAN decoder to reconstruct the image?

Why do you use MAE to fill the missing parts? Why not just use the transformer from VQGAN to generate the missing parts directly? Is the MAE part necessary?

How do you assign probabilities to visual tokens in L111?

Does the prediction of z^{\hat}_k related to z^{\hat}_{k-1} in the eqn under L113? Does each z^{\hat}_k are predicted only based on x and m?

In the eqn right above L119, is m a mask of the part needs to be filled?

If the authors can address all the above questions well, I think this is a good paper to be accepted.



**Limitations:**

Some model details are missing. For example, the model details of MAE and VQGAN as mentioned above.

**Strengths And Weaknesses:**

(+) The authors propose a smart way to solve various image tasks by treating them as an inpainting task. This paper is very impressive. The idea is novel and the paper is well-written. The results are convincing.

(-) Some method details and implementation details can be added to improve this paper.

(-) This paper did not compare with the standard methods for image segmentation, colorization, object detection, etc. For example, Mask-RCNN [1], Mask2Former[2], [3] and so on. The authors should add comparison with these standard methods and the state-of-the-art methods in each domain.

[1] Mask R-CNN
[2 ] Masked-attention Mask Transformer for Universal Image Segmentation
[3] Real-Time User-Guided Image Colorization with Learned Deep Priors

---

> ### Author Response · Authors · 2022-08-02
> **Response to reviewer xQdx**
>
> Thank you for your positive comments and suggestions. We address your comments below.
>
> Q: **Comparison to baselines**. Thank you for suggesting this, we include comparisons to finetuning and state-of-the-art classic few-shot approaches. But note that classic few-shot models should be considered an upper bound of our approach as they are trained on large amounts of clean and annotated (baseclass) data and were designed for a specific task. We discuss this in the shared comments and in the Supplementary Material Section 6.1.
>
> Q: **How did you combine the MAE and VQGAN? During training, did you train VQGAN first? And then use MAE to reconstruct the visual tokens? During inference, do you encode the whole combined image using VQGAN first? And then use MAE to fill the missing part (visual tokens instead of pixels)? And then use VQGAN decoder to reconstruct the image?** MAE-VQGAN utilizes an ImageNet pre-trained VQGAN codebook [1] which is fixed during training and inference, we clarify it in the revised manuscript (lines 110-111). As you mentioned, instead of directly regressing pixels, the MAE-VQGAN model predicts visual tokens which can then be decoded into pixels. However, the input to MAE-VQGAN is pixels and not visual tokens (unlike in the original VQGAN model).
>
> [1] https://github.com/CompVis/taming-transformers
>
>
> Q: **How do you assign probabilities to visual tokens in L111?** The MAE-VQGAN model outputs a distribution over the visual tokens via a softmax layer. This is unlike MAE which is trained to regress pixels. We clarify this in the revised manuscript (line 112).
>
> Q: **Why do you use MAE to fill the missing parts? Why not just use the transformer from VQGAN to generate the missing parts directly? Is the MAE part necessary?** We found that using the full VQGAN model leads to worse performance (see Tables 1 and 2). We believe the main reason is sequential decoding, which is one of the assumptions of VQGAN (we discuss this in the original submission, lines 204-205 and 222-223).
>
> Q: **Does the prediction of z^{\hat}k related to z^{\hat}{k-1} in the eqn under L113? Does each z^{\hat}_k are predicted only based on x and m?** Yes, here we do not assume an order, e.g, we do  not condition the z^{\hat}k on the prediction of z^{\hat}{k-1}. However, exploring different decoding strategies is a promising research avenue and we leave this for future work.
>
> Q: **In the eqn right above L119, is m a mask of the part needs to be filled?** Yes. We mention this in the original submission (Sec 3.1 line 118-119).

---

> > ### Comment · Reviewer_xQdx · 2022-08-09
> > **More questions**
> >
> > Thanks for your reply. However, there are still some details missing.
> >
> > 1. During testing, did you test tasks that were not seen during training? I do not mean new testing examples, but instead new tasks, such as converting RGB images to BGR.
> >
> > 2. The training and inference process of MAE-VQGAN is still not clear. Could you explain how you encode the support images + query? Did you combine them first and then resize them to 224 $\times$ 244 or resize each image to 224 $\times$ 224 and then combine them?
> >
> > If you combine them first and then resize them to 224 $\times$ 244, then why the VQGAN pre-trained on ImageNet can deal with such images that are quite different from the images in Imagenet? And did you give a mask of the query image, which means the VQGAN tasks (resized support images + query image + query mask) as input and generate the tokens?
> >
> > If you resize each image to 224 $\times$ 224 and then combine them? How could you deal with more support examples? In MAE, they use ViT (max sequence length=1024). If you have > 2 examples (4 support images + 1 query image, each generating 16 $\times$ 16 tokens), the token sequence is too long to feed them into ViT.
> >
> > 3. What do you mean 'However, the input to MAE-VQGAN is pixels and not visual tokens (unlike in the original VQGAN model).'?
> > Could you explain clearly the input and output of the VQGAN part and the MAE part? When you train MAE, did you feed in the combined image grid or every single image?
> >
> > 4. How did you generate the mask of the query? If the grid is 3 $\times$ 2, the mask might be encoded into a half token, such as 1.5 tokens. How will you decode that? Is the basic block of a binary mask corresponding to a pixel or a patch of ViT? During training, do you calculate the loss over the whole picture or the masked part?
> >
> > There are still many details missing. I think adding algorithm figures and detailed explanations of training and inference, or providing the code are necessary.

---

> > > ### Author Response · Authors · 2022-08-09
> > > **R2R**
> > >
> > > Thank you for the questions and suggestions. As the discussion phase ends today, we will revise the paper to include the missing technical details and the training/inference figure suggested after the discussion period. Additionally, the code will be made publicly available.
> > >
> > > Q: **During testing, did you test tasks that were not seen during training, some new image tasks?** We evaluate our model on a variety of tasks that are unlikely to have been seen during training. For example, our Synthetic Data Study (Section 4.3) presents a family of 6 previously unseen tasks like Color, Shape, and Size change. In addition, in the Supplementary Material (Section 12), we include tasks like image contrast change, text font completion, and transforming an image to grayscale.
> > >
> > > Q: **“The training and inference process of MAE-VQGAN is still not clear.”, “I think adding algorithm figures and detailed explanations of training and inference are necessary”**. During training, MAE-VQGAN model is trained on masked auto-encoding similarly to standard MAE except for a single change. Whereas in MAE the model is trained to complete raw pixels, MAE-VQGAN decoder is trained to predict VQGAN codebook visual tokens (Section 3.1). Assuming we use a ViT model that patchifies a 224x224 image to 14x14 patches, MAE predicts 16x16x3 values per patch, and MAE-VQGAN predicts 1024 values per patch that represent a distribution over the pre-trained VQGAN codebook.
> > >
> > > In inference, we feed Visual Prompts to MAE-VQGAN that predicts VQGAN visual tokens (Section 3.2). Decoding the predicted visual tokens to pixels is done using the VQGAN codebook decoder (See Section 3.1, line 115).
> > >
> > > Q: **Could you provide more explanation of how you encode the support images + query? Did you combine them first and then resize to 224x244?** Yes, to construct the Visual Prompt (inference only) we combine the support images + query and then resize them to 224x224. Before combining everything, we add a white border between the images, to create a separation between images in the grid.
> > >
> > > Q: **If you combine them first and resize to 224*244, then why the VQGAN pre-trained on ImageNet can deal with such images that are quite different from the images in Imagenet?** We empirically observe that the reconstruction quality of the VQGAN encoder-decoder is plausible and does not suffer from a large domain shift (e.g., it can reasonably reconstruct grid images with black and white segmentation masks). However, as we acknowledge in the Limitations (see lines 253-255), the ImageNet pretrained VQGAN codebook might suffer from domain shift. To alleviate this difficulty, it is possible to train a new codebook on a larger and more diverse dataset.
> > >
> > > Q: **Could you explain clearly the input and output of the VQGAN part and the MAE part?** Like in a standard MAE, the input to MAE-VQGAN is an image and an associated mask. The mask corresponds to the set of patches that comprise the image. Differently from MAE, the output of MAE-VQGAN are VQGAN codebook visual tokens. For example, based on a ViT model that patchifies a 224x224 image to 14x14 patches, MAE predicts 16x16x3 values per patch, and MAE-VQGAN predicts 1024 values per patch that represent a distribution over the pre-trained VQGAN codebook.
> > >
> > > During training, we use the VQGAN codebook encoder to map an image to a list of visual-tokens (each 224x224 image translates to 14x14 codebook indices) to provide ground-truth for MAE-VQGAN predicted visual tokens.
> > >
> > > During inference, to decode the visual tokens predicted by MAE-VQGAN to pixels, we take the argmax and then use the VQGAN codebook decoder that translates the set of 14x14 tokens to a 224x224x3 image.
> > >
> > > Q: **When you train MAE, did you feed in the combined image grid or every single image?** We only construct the image grid (Visual Prompt) during inference time. In training, our model is trained to predict the visual tokens of randomly masked regions.
> > >
> > > Q: **What do you mean 'However, the input to MAE-VQGAN is pixels and not visual tokens (unlike in the original VQGAN model).'?** MAE-VQGAN relies on a standard ViT backbone like MAE. This is different from the original VQGAN transformer model that first encodes the image using the VQGAN codebook encoder before processing it.
> > >
> > > Q: **How did you generate the mask of the query? If the grid is 3x2, the mask might be encoded into a half token, such as 1.5 tokens. How will you decode that? Is the basic block of a binary mask corresponding to a pixel or a patch of ViT?** The basic block of the binary mask corresponds to a patch of ViT. If the size is not divisible by the number of tokens (e.g, 1.5 tokens like you mentioned) we round up and mask a larger region (e.g, 2 tokens). We plan to clarify it in the Supplementary.
> > >
> > > Q: **During training, do you calculate the loss over the whole picture or the masked part?** As in MAE, the loss is calculated only on the masked part.

---

### Official Review · Reviewer_WRn2 · 2022-07-04

**Rating:** 6
**Confidence:** 4
**Ethics Flag:** Yes
**Soundness:** 2 fair
**Presentation:** 3 good
**Contribution:** 3 good

**Summary:**

The paper proposed a dataset constructed of figures from computer vision and machine learning arXiv papers filtered for grid-style figures. This allows training an image in-painting model on a large variety of input-output examples (contained in one image). The model can then be used to solve several popular tasks by “prompting” in the sense that an image is constructed as a grid of input-output examples with the last output being masked for in-painting. The model prediction can then be interpreted as a task prediction.
The paper contains a broad evaluation of different architectures, tasks, and prompts and shows that in many arrangements the final model produces good results when trained on this dataset vs. ImageNet.


**Questions:**

The questions here directly reflect the weaknesses above.
1. What is the definition of few-shot learning in this paper and how does it relate to the existing approaches of few-shot learning?
2. How can we make sure that the evaluation of the model uses unseen categories/tasks/images if the dataset is sourced from results of other papers?
3. How was the dataset collected? What does it contain? Were individual licenses taken into account?
4. Does the model pick up a bias in artificially predicting imprecise results to emulate predictions?

**Ethics Review Area:**

["Privacy and Security (e.g., consent)", "Legal Compliance (e.g., GDPR, copyright, terms of use)"]

**Limitations:**

The paper contains a section on technical limitations of the work but does not discuss the societal impact of the work nor the dataset. A datasheet is strongly encouraged as well as a more clear description on how the dataset was collected and filtered.

**Strengths And Weaknesses:**

Strengths

The paper shows a straight-forward way to train an inpainting model to be used for visual prompting at inference time. This idea is quite interesting and the trained model shows reasonable performance on several tasks.

The paper evaluates several architectures, training datasets, prompting schemes and tasks, and analyzes all major design choices to show that the final model performs best.

The proposed dataset of figures from arxiv papers and could be of broader interest to the research community.


Weaknesses

Few-shot learning: The paper claims that models trained like that are few-shot learners. However, this statement is based on a very stretched definition of few-shot learning. The most common example of few-shot learning in computer vision is classification, where a model is trained on a dataset of training classes with images. During test time, the model is shown a few annotated examples of a previously unseen class and has to produce a classifier that can recognize this class. The important difference here is that in this setup (and others such as few-shot detection, semantic segmentation, etc.) is that while the task remains the same, the model is presented with unseen categories at test time. In this paper the setup is quite different, as the evaluation does not test unseen categories. Without a proper analysis of the dataset (which is missing) it is unclear which categories, tasks and even images have been seen during training. As most of the tasks that are evaluated in the paper are highly popular computer vision tasks with hundreds of papers on arxiv (segmentation, colorization, detection, inpainting, edge detection) there is a high chance that the model simply recognizes the task from the prompt instead of learning a new category or a new task from the given examples.

Dataset: There is very little information about the collected dataset in the paper and appendix which makes it not only difficult to assess the findings of the paper but also leaves many questions on the dataset itself. I strongly suggest creating a “datasheet for datasets” for this dataset. For example: it is not clear if the license of the arxiv papers has been taken into account before extracting the figures. It is unclear if copyright and attributions are retained in the dataset. By nature, the dataset will consist of many validation/testing images from popular datasets, thus training a model on this data will likely skew its performance when evaluated on the val/test sets for downstream task performance.

Learning Signal: This point is related to the content of the dataset. Figures in papers often compare the results of the method with the ground truth and other methods. With this prompting approach it could be possible that a model learns to predict imprecise results from the fact that papers most often show predictions. It could be interesting to investigate how much this affects the method. Potential ideas could be adding labels (e.g. “GT”, “pred”, “ours” to the columns).

---

> ### Author Response · Authors · 2022-08-02
> **Response to reviewer WRn2**
>
> Thank you for the constructive review and suggestions. We address your comments below.
>
> Q: **I strongly suggest creating a “datasheet for datasets” for this dataset.** Thank you for the suggestion. We’ve uploaded a new revision and included a datasheet in Supplementary Material Section 7.
>
> Q: **What is the definition of few-shot learning in this paper and how does it relate to the existing approaches of few-shot learning?** We use the term "Few-Shot" as used in the NLP community and in particular as defined in the seminal paper “Language Models are Few-Shot Learners” [3]. In this relatively new setting, the model’s input contains very few input-output examples (in our work, usually less than 2) and new input. The goal of the model is then to deduce the rule from the examples and apply it to the new input. This setting is different from the classic computer vision Few-Shot setting as it does not assume a specific task and the model is not pre-trained on large amounts of base class labeled data. In the updated version, we do have a comparison to classic Few-Shot methods, but we note that this should be treated as an upper limit.
> We discussed the two uses of the term “Few-Shot” in the original submission (see introduction lines 41-43 and related work lines 79-83).
>
> Q: **How can we make sure that the evaluation of the model uses unseen categories/tasks/images if the dataset is sourced from the results of other papers?** As discussed in the paper (see lines 41-43 and lines 79-83), we follow the definition of Few-Shot as presented in [1], instead of the classic computer vision definition (that is usually restricted to a single task and utilizes base class training). Therefore, we do not distinguish between categories and tasks.
> We analyzed the potential overlap between the Figures dataset and the test data and found no significant overlap (see Supplementary Material, “Train-test overlap”, lines 453-470). Following your question, we double-checked this again. Specifically, to evaluate the overlap, for each image of 100 random Pascal 5i test images we computed the 5 nearest neighbors in the Figures datasets using CLIP embeddings. Out of the 100 images, we found only a single one contained in a Figure image, and this image did not have any associated ground-truth annotation. Therefore, we conclude that even if there are potential overlaps they are likely very small and insignificant. We discuss this in the revised Suppl. Sec 7 “Composition”.
>
> Q: **How was the dataset collected? What does it contain?** Latex source files were originally collected by the Arxiv team between the years 2010 to 2022. From this data, we automatically extracted figures containing images (as discussed in Section 3.3). By manually annotating a random sample of 500 images, we estimate that the majority of the images (84%) contain grid-like images, where 60% of the images contain some annotation embedded in the image, like text, arrows, heatmap, bounding box, etc.
> We described the exact data collection process in the original submission Section 3.3 and Suppl Section 6. We also included the distribution of image types in the Suppl Table 5. Following your suggestion, we’ve added a datasheet for the dataset in the revised Supplementary Material.
>
> Q:  **(your) dataset will consist of many validation/testing images from popular datasets, thus training a model on this data will likely skew its performance when evaluated on the val/test sets for downstream task performance.** Thank you, this is a good point. To evaluate this potential skew in performance, we analyzed our model (MAE-VQGAN) on the colorization task on ImageNet train vs. validation sets (better performance on the validation set would indicate a skew). Specifically, we randomly sampled two sets of 1000 images from train and validation and compared their performance. We observe an insignificant difference in LPIPS metric (0.39 for train vs. 0.40 for val) which suggests there is no evidence for this particular bias. We plan to add this to the paper.

---

> > ### Author Response · Authors · 2022-08-02
> > **Response to reviewer WRn2 (cont.)**
> >
> > Q: **With this prompting approach it could be possible that a model learns to predict imprecise results from the fact that papers most often show predictions. Does the model pick up a bias in artificially predicting imprecise results to emulate predictions?** We agree that potential bias towards imprecise results may arise and following your feedback we now discuss this in the revised Supplementary Material datasheet under Composition.
> > Generally speaking, the goal of our paper is not to present a practical solution for any specific task, but rather to demonstrate how our model can be prompted at test time to solve various tasks **without any training**, only by pre-training on noisy, unlabeled data. Surprisingly, as presented in the synthetic data study (Section 4.3) and in the Supplementary (Section 12), our model shows the ability to generalize to new tasks as well.
> >
> > Q: **Were licenses taken into account?** Yes, we have considered Arxiv licenses (https://arxiv.org/help/license). As you suggested, we include the full information in a dataset sheet in the revised Suppl. Section 7 “Distribution”.

---

> > > ### Comment · Reviewer_WRn2 · 2022-08-08
> > > **Thank you for the clarifications**
> > >
> > > I have read the other reviewes, the responses and the updated paper. The additional experiments on val vs. train bias and the few-shot baselines are valuable.
> > >
> > > The provided explanations were quite helpful and I did not find any major unadressed points in the other reviews.
> > > Thus, I am raising my rating to recommend acceptance.

---

### Official Review · Reviewer_HDgS · 2022-07-09

**Rating:** 6
**Confidence:** 4
**Soundness:** 2 fair
**Presentation:** 3 good
**Contribution:** 2 fair

**Summary:**

The paper aims to generalize the idea of natural language prompting to the vision domain. They propose visual prompting by image inpainting. Given task examples (image, target) and a image query, they construct a new “grid-like image” (i.e., visual prompt) and the inpainting model predicts the masked region such that it is consistent with the example(s). Given that it was trained on the right data, the inpainting model can perform prompting in various vision tasks. One key assumption is that the tasks can be defined as an image-to-image translation problem. To train the in-painting model, they collect a dataset of 88k figures, which is critical for performance.

**Questions:**

* How does the method compare to baseline methods (state-of-the-art, fine-tuning, zero shot)? In fact, what is the practical utility of this prompting method? The benefit of prompting is the ability to generalize to a new task *without additional training*. However, the proposed method requires having to train a whole new model which essentially contradicts the value of prompting. If the prompting method only works on the proposed pre-trained model, how useful/general is this pre-trained model? Does the proposed pre-training + prompting method outperform training from scratch using the same dataset/model with a task-specific state-of-the-art objective? If not, why do we have to use this prompting method?

* I’m confused about how the datasets (ImageNet, Figures) are processed during pre-training. Are they concatenated as a grid-like image and processed as input or is only a single image used as input? From what I understand, during pre-training, a single image is used as input and a random region is masked, and the model is trained to reconstruct the image. I believe this is inefficient. For prompting, closely matching the pre-trained and downstream datasets is key. Therefore, rather than collecting a whole new dataset (Figures) and do random region masking, can’t you reformulate existing datasets (e.g. ImageNet) into a grid-like structure and pre-train the model? Have you tried this? How does this performance compare to having the models pre-trained on the Figures Dataset with random masking? This would further reduce the domain gap between pre-training and downstream tasks.

**Limitations:**

I think the key limitations of this work are domain gap between pre-trained and downstream task,  scalability (the pre-trained model specifically requires figure-like training images), and possibly performance (does this method outperform training from scratch using state-of-the-art?). While I really like the pure vision approach for prompting, these limitations make me question the practical utility of this work.

**Strengths And Weaknesses:**

Strengths
* It is very interesting to see that in-context learning in NLP works for images, too. While the set-up is extremely simple, the paper shows that pre-trained task of image in-painting is flexible enough to cover a wide variety of vision tasks.
* It is one of the first papers that shows that in-context learning works for vision. While other works incorporate vision into a language modeling task or focus on using language prompts, I like that this paper takes a “purely vision” approach. One value of this paper is showing that prompting can be effective for vision *without* using language pre-training tasks or language prompts.
* Overall, the paper is well-written and easy to follow.

Weaknesses
* Having to collect a whole new dataset (Figures) specific to this pre-trained task lacks scalability, when having a sufficiently large training set is crucial for creating a high-quality pre-trained model.
* In fact, how the authors perform pre-training seems inefficient. For prompting, closely matching the pre-trained and downstream tasks is key. While their prompt for the downstream tasks is “grid-like images” with support examples and a query image, the pre-training task is in-painting with random region masking. This creates an unnecessary domain gap between pre-training and downstream tasks. Can’t we just reformulate existing datasets (e.g. ImageNet) into a grid-like structure and use them for pre-training?
* The paper lacks baseline comparisons. For Table 1, how does the method compare to state-of-the-art, fine-tuning, and zero shot? It is very difficult to assess the value of the proposed method without any comparisons. Essentially, prompting is a method that allows *no additional training*. This paper requires *pre-training a whole new model* specific to the proposed prompting method which contradicts the value of prompting. Have you tried using the prompt on any off-the-shelf pre-trained in-painting model (i.e., zero shot)? Or such a pre-trained model doesn't exist for the proposed prompting method?
* For colorization, you should measure LPIPS [1] instead of MSE. MSE cannot accurately measure the perceptual similarity between colored output and the target.

[1] The Unreasonable Effectiveness of Deep Features as a Perceptual Metric, Zhang et al., CVPR 2018.

---

> ### Author Response · Authors · 2022-08-02
> **Response to reviewer HDgS**
>
> Thank you for the constructive review and suggestions. We address your comments below.
>
>
> Q: **how the datasets (ImageNet, Figures) are processed during pre-training. Are they concatenated as a grid-like image and processed as input or is only a single image used as input?** There is no pre-processing of images. During pre-training, we use the original images as is, where each image is considered as a single input to the model.
>
> Q: **This paper requires pre-training a whole new model specific to the proposed prompting method which contradicts the value of prompting.** Our main focus was to present a proof of concept that shows it is possible to visually prompt Image Inpainting models that were trained on noisy, unlabeled data. Specifically, we show how we can pre-train a network once, then prompt it to perform reasonably well on a very wide variety of tasks (both established as well as novel). The fact that this is possible is surprising and scientifically interesting, as was also mentioned by other reviewers (v9Mo, WRn2).
>
> Q: **Can’t we just reformulate existing datasets (e.g. ImageNet) into a grid-like structure and use them for pre-training?** Indeed, it is possible to reformulate existing datasets into a grid-like structure and train a single model on all datasets and tasks. However, this requires using explicit annotations in each dataset. Moreover, this approach cannot scale to the huge number of tasks (hundreds) and datasets (thousands) that appeared in computer vision papers over the years, as each task requires manual reformulation.
>
> More broadly, our goal in this work is not to present a practical algorithm for any specific computer vision task, but rather to demonstrate how a model can be adapted at test time to various tasks **without any training**, only by learning from unlabeled and unstructured data. As presented in the synthetic data study (Section 4.3) and in the Supplementary (Section 12), our model shows the ability to generalize to new tasks outside of the training distribution. We also show that this approach is surprisingly scalable when additional unlabeled (natural) images are blended into the proposed Figures dataset (Section 4.4).
>
> Q: **Having to collect a whole new dataset (Figures) specific to this pre-trained task lacks scalability.** One of our main findings in this work is that blending the Figures dataset with unlabeled ImageNet leads to significantly better performance (Section 4.4 - “Dataset effect”). As unlabeled images can be obtained easily, we believe that this approach is scalable.
>
> To further demonstrate that our approach scales when using additional large amounts of unlabeled data, we train MAE-VQGAN with a longer schedule (10k epochs instead of 1k) on a combined dataset of Figures and ImageNet. The resulting model achieves mIOU of 34 compared to 31 of Figures only, which is more than a 10% improvement.  We plan to update the paper to reflect this.
>
>
> Q: **Comparisons to baselines.** Thank you for suggesting this, we include comparisons to finetuning and classic few-shot approaches and discuss this in the shared comments and in the Supplementary Material Section 6.1. But note that state-of-the-art classic few-shot models should be considered an upper bound of our approach as they are trained on large amounts of clean and annotated (baseclass) data and were designed for a specific task.
>
> Q: **For colorization, you should measure LPIPS.** Thank you for the suggestion. We added the LPIPS metric in Table 1 of the revised version.
>
> Q: **Have you tried using the prompt on any off-the-shelf pre-trained in-painting model (i.e., zero shot)? Or such a pre-trained model doesn't exist for the proposed prompting method?** Yes, we have tried to use the prompt on off-the-shelf models. For example, we used  pre-trained VQGAN model, which is a state-of-the-art transformer-based inpainting model (see results in Table 1 and 2). However, we found that it performs poorly. This is mainly because inpainting models are intended to complete natural images which are different than our proposed Visual Prompts.

---

> > ### Comment · Reviewer_HDgS · 2022-08-09
> > **Response to the rebuttal**
> >
> > I have read the rebuttal and the other reviews. The answers were generally helpful, but I do have one unaddressed point:
> >
> > > Can’t you reformulate existing datasets (e.g. ImageNet) into a grid-like structure and pre-train the model? Have you tried this? How does this performance compare to having the models pre-trained on the Figures Dataset with random masking?
> >
> > Reformulating ImageNet images into a grid-like structure and pre-training the in-painting model does not need any annotations, but the authors did not provide the comparison. In general, I am raising my rating to a borderline accept.

---

> > > ### Author Response · Authors · 2022-08-09
> > > **R2R**
> > >
> > > Thank you for the comment and for raising your score.
> > >
> > > > Reformulating ImageNet images into a grid-like structure and pre-training the in-painting model does not need any annotations
> > >
> > > Can you elaborate on the exact way you propose to reformulate ImageNet images into a grid-like structure without annotations?
> > >
> > > For example, choosing random ImageNet images then placing them in corresponding grid cells is one way. However, since there is no rule to infer one grid cell from the other, the model will learn to complete each cell individually while ignoring the rest of the image. Therefore the model will not learn completions that are relevant to the tasks we discuss in the paper (e.g segmentation/colorization).

---

> > > > ### Comment · Reviewer_HDgS · 2022-08-09
> > > > **Re: R2R**
> > > >
> > > > Thank you for your response. I understand the author's comment on requiring annotations for ImageNet training. I still wonder how the model would perform by reformulating existing datasets into a grid-like structure using additional annotations, as prompting works best by minimizing the gap between pre-trained and downstream tasks. This can come as future work. I will modify my final score to a weak accept.

---

### Official Review · Reviewer_v9Mo · 2022-07-10

**Rating:** 7
**Confidence:** 3
**Soundness:** 3 good
**Presentation:** 3 good
**Contribution:** 4 excellent

**Summary:**

This paper proposes a novel approach to use a pre-trained visual model to various downstream tasks via “prompting”. In other words,  it can be viewed as a one-shot or few-shot learner but without the need for fine-tuning. Similar to NLP systems, they propose to use image inpainting as the pre-training task. To prompt for a downstream task, the method concatenates the exemplar pair images and the input image into a larger “visual prompt image”. Notably, they demonstrate that the dataset that the visual model is pre-trained on is important. Specifically, they collected images of paper figures from arXiv papers. Empirically, they demonstrate their approach to various downstream tasks, including foreground segmentation, single object detection, colorization.

**Questions:**

See #2, #3, #4 from above.

**Limitations:**

The paper contains a limitation section and adequately discussed the shortcomings of the approach. Specifically, the inherit ambiguity when prompting from a single image for a task.

**Strengths And Weaknesses:**

# Strengths
The proposed approach is novel and interesting. The authors proposed a simple (in a good way) approach to enable prompting on visual models with many analogues to NLP. Personally, I find the insight to training on arXiv paper figures intuitive and creative. Overall, the writing is organized and clear. Finally, the authors also promise to release the data and code, which would benefit the community.
# Weaknesses
#### 1. I believe the paper would benefit from a comparison with a one-shot baseline. While the proposed approach does not require fine-tuning, it would be interesting to see how it competes.
#### 2. The prompt engineering details are unclear. I was only able to get a vague idea on how to construct the visual prompt. Is it possible to document them more precisely? Additionally, how are different image sizes being handled? Are the images being resized?
#### 3. Missing training details? Specifically, I am wondering if the VQGAN is pre-trained? Or only trained on the 88,635 images from the Computer Vision Figures dataset.
#### 4. The authors have stated that they repeated the experiment with four random seed (line. 194). It would be great to report the standard deviation on the quantitative results. Also, I wonder how sensitive the approach is to the prompted image. For example, would the approach still work if the cat, in Fig.3, is prompted by an image of a white cat that's outdoor?

# Misc.
- Line 173: "224x224" --> "224 \times 224"

---

> ### Author Response · Authors · 2022-08-02
> **Response to reviewer v9Mo**
>
> Thank you for the positive review and suggestions. We address your comments below.
>
> Q: **how sensitive is the approach to the prompted image?** While there might be differences, as long as the supports are not degenerate (e.g, for segmentation - the support mask is all foreground or all background), the overall quality of the result remains similar. To demonstrate this, we include additional prediction examples where the support images change while holding the query image fixed (See the Suppl index.html, Section 13, “Support examples effect”).
>
> Q: **Report the standard deviation on the quantitative results.** ​​Thank you for the suggestion. We’ve included a table with the mean and standard deviations in the revised Supplementary Material (Section 6.1, Table 6).
>
> Q: **how to construct the visual prompt, how are different image sizes being handled? Are the images being resized?** Yes, the images are resized. The grid has a fixed size (the image size) of 224x224 pixels, and thus to construct the Visual Prompt the support pairs and image query are first resized and then placed in a grid. We clarify this in the revised manuscript (lines 126-127).
>
> Q: **is the VQGAN pre-trained? Or only trained on the 88,635 images from the Computer Vision Figures dataset?** Yes, the VQGAN is pre-trained. Specifically, we use an ImageNet pre-trained VQGAN codebook [3]. We clarify this in the revision (lines 110-112).
>
> Q: **Comparison with baselines**. Thank you for the suggestion, we’ve included additional comparisons, please see the shared comment or the updated Supplementary materials for the full information.
>
>
> [3] https://github.com/CompVis/taming-transformers

---

> > ### Comment · Reviewer_v9Mo · 2022-08-08
> > **Thanks for the response**
> >
> > I have read the rebuttal and the other reviews. I am maintaining my original evaluation of accept.

---

> > ### Comment · Reviewer_xQdx · 2022-08-09
> > **Question**
> >
> > I have the same question on how the visual prompt is constructed.
> >
> > For example, you have two support examples (for example, each example has an RGB image and a segmentation map), and a query image. You said you create an image grid of (n + 1) * 2 cells (L125).
> >
> > Did you resize this large combined image grid to 224*224? And then feed it into the pre-trained VQGAN?
> >
> > If so, it is questionable why the VQGAN trained on ImageNet (a single image) can encode the combined image grid well. The combined image grid should be very different from images in ImageNet.
> >
> > Please let me know your answers.

---

### Review · Ethics_Reviewer_4BDU · 2022-08-02

**Recommendation:**

As stated above, the authors should clarify their compliance with the different licenses made available to those who upload content to arXiv.

**Ethical Issues:**

Yes

**Ethics Review:**

Reviewer WRn2 expressed a concern that this paper did not provide a sufficient level of detail on the nature of the arXiv figure dataset. In particular, this reviewer drew attention to the lack of clarity around licensing information for the individual arXiv papers and also suggested that the authors provide a datasheet documenting the collection process. The authors have since provided a datasheet as supplementary material, but the issue around individual arXiv paper licenses remains. The 'arXiv license' they link to in their datasheet is not the only license available to authors who publish on arXiv; did the authors restrict their data downloads to such papers, or are they conflating this license with the other license types? They should clarify this.
The datasheet claims that there are no people in the dataset, but Figures 2 and 6 of the paper (derivative figures from other papers) do contain images of people.
The authors have not discussed the societal impact of their proposal, for example the potential misuse of image manipulation technology. One could imagine prompting the inpainting model with unsavory images to realistically falsify events or generate harmful images. Do the authors have a proposal for mitigating negative outcomes such as this?

---

### Author Response · Authors · 2022-08-02
**General Response**

We thank the reviewers for their insightful comments, which we incorporated into a new revision of the paper. The reviewers found our approach to Visual Prompting “interesting” (HDgS, v9Mo, WRn2), “novel” (v9Mo, xQdx), and “creative” (v9Mo) and identified it as one of the first works that show “prompting is effective for vision without using language” (WRn2). Additionally, the reviewers noted that the approach is “flexible enough to cover a wide variety of vision tasks” (HDgS, xQdx, WRn2).

The reviewers requested comparisons to finetuning or few-shot baselines. We follow their suggestion and add finetuning baselines that utilize K={1, 4, 16} training examples for each target class. We also include the results of FWB [1] and CyCTR [2], classic state-of-the-art 1-shot models, which we view as an upper bound of our approach. These approaches utilize a fully labeled base classes train set (2086 to 5883 on different Pascal 5i splits). Additionally, both architectures were designed for the foreground segmentation task (e.g, they operate in higher resolution).

The results indicate that the Visual Prompting results of MAE-VQGAN trained on Figures are significantly superior to standard finetuning baselines of MAEs pretrained on unlabeled ImageNet. As mentioned in our original submission (lines 79-93), classic few-shot approaches pretrain on a large, clean,  tagged base classes dataset and utilize task-specific architectures. Thus, FWB [1] and CyCTR [2] unsurprisingly outperform Visual Prompting, but Visual Prompting is more general as it can be applied to many novel tasks without any fine-tuning. We also include these results in the revised Supplementary Material Section 6.1.



| Pretraining                                | #Labeled Examples                   | #shots | Model         | Split1 | Split2 | Split3 | Split4 |
|--------------------------------------------|--------------------------------------|---------|---------------|--------|--------|--------|--------|
| Unlabeled Figures Dataset                 | 1                                    | 1       | Visual Prompting MAE-VQGAN     | 27.8   | 30.4   | 26.1   | 24.3   |
|  Unlabeled ImageNet.                         | 1                                    | 1       |  Finetune MAE | 11.1   | 13.4   | 13.0   | 12.3   |
|  Unlabeled ImageNet                          | 4                                    | 4       |  Finetune MAE | 12.9   | 15.8   | 14.3   | 15.0   |
|  Unlabeled ImageNet                          | 16                                   | 16      | Finetune MAE | 13.7   | 16.1   | 16.8   | 17.1   |
| **Labeled Pascal 5i (segmentation masks)** | 2086-5883 (different per each split) | 1       | FWB [1]       |  51.3  | 64.5   | 56.7   | 52.2   |
| **Labeled Pascal 5i (segmentation masks)** | 2086-5883 (different per each split) | 1| CyCTR [2]     |  67.2  | 71.1   | 57.6   | 59.0   |




[1] Nguyen, Khoi, and Sinisa Todorovic. "Feature weighting and boosting for few-shot segmentation." ICCV 2019.

[2] Zhang et al. "Few-shot segmentation via cycle-consistent transformer." NeurIPS 2021.

---

### Meta-Review · Area_Chair_1iEz · 2022-08-25

**Recommendation:** Accept
**Confidence:** Certain

**Metareview:**

The paper discusses a way to use pre-trained models for downstream tasks. Reviewers generally appreciated the paper but had questions regarding baselines, details, dataset, etc. The rebuttal addressed most of these concerns prompting the reviewers to raise their recommendation. However some questions remained (e.g., https://openreview.net/forum?id=o4uFFg9_TpV&noteId=wPCrVV96hE). AC concurs with the unanimous reviewer recommendation.

**Award:**

No

---

### Decision · Program_Chairs · 2022-09-14

Accept